# Agglomerate Size Evolution in Solid Propellant Combustion under High Pressure

**Songchen Yue** [1], **Lu Liu** [1], **Huan Liu** [1], **Yanfeng Jiang** [1], **Peijin Liu** [1], **Aimin Pang** [2,*], **Guangxue Zhang** [3] **and Wen Ao** [1,*]

1   Science and Technology on Combustion, Internal Flow and Thermo-Structure Laboratory, Northwestern Polytechnical University, Xi'an 710072, China; yue_songchen@163.com (S.Y.); 2022100051@mail.nwpu.edu.cn (L.L.); liuhuanspace@mail.nwpu.edu.cn (H.L.); 13559781518@163.com (Y.J.); liupj@nwpu.edu.cn (P.L.)
2   Science and Technology on Aerospace Chemical Power Laboratory, Xiangyang 441003, China
3   Institute of Energy Engineering, China Jiliang University, Hangzhou 310018, China; zhangguangxue@cjlu.edu.cn
*   Correspondence: ppam@tom.com (A.P.); aw@nwpu.edu.cn (W.A.)

**Abstract:** Solid propellant combustion and flow are significantly affected by condensed combustion products (CCPs) in solid rocket motors. A new aluminum agglomeration model is established using the discrete element method, considering the burning rate and formulation of the propellant. Combining the aluminum combustion and alumina deposition model, an analytical model of the evolution of CCPs is proposed, capable of predicting the particle-size distribution of completely burned CCPs. The CCPs near and away from the propellant burning surface are collected by a special quench vessel under 6~10 MPa, to verify the applicability of the CCP evolution model. Experimental results show that the predicted error of the proposed CCP evolution model is less than 8.5%. Results are expected to help develop better analytical tools for the combustion of solid propellants and solid rocket motors.

**Keywords:** aluminized solid propellant; condensed combustion products; agglomeration; combustion; particle size prediction model





## 1. Introduction

In recent years, aluminized solid propellants have received considerable attention, due to several advantages, including a higher propellant energy density, a higher combustion enthalpy, and a higher level of stability [1–3]. As well as the combustion of the propellant, condensed combustion products (CCPs) from aluminum also play a role in determining the energy release of a solid rocket motor [4–6]. There are a number of characteristics of motor systems that are affected by these products, including propellant energy release, specific impulse loss, slag deposition, and combustion instability [7–9]. To accurately estimate these effects, it is necessary to establish a method for predicting the CCP particles during the combustion of solid composite propellants.

Aluminum is extensively studied for its agglomeration and combustion characteristics in solid composite propellants. There are three categories of theoretical models which address the agglomeration process. A majority of early work focused on empirical models of experimental fitting, such as the studies of Salita [10], Hermsen [11], and Beckstead [12]. Using these models, the particle size and agglomeration ratio can be calculated easily, based on propellant formulation or pressure. Models based on empirical data are difficult to improve because they do not take into account the physical properties of agglomerations. As well as this, they cannot provide results for the size distribution of agglomerates, which is crucial when examining the erosion of chamber walls or nozzles.

The pocket model, by contrast, is more appealing, since the pocket model is simpler and its description of physical properties is more reasonable. The basic idea of the pocket model is that all aluminum particles agglomerate into one large mass, and these aluminum particles are contained in pockets formed by the ammonium perchlorate (AP) particles. The models of Cohen [13] and Grigorev [14] are widely known in all pocket models. The fraction of agglomeration depends on the amount of molten aluminum in the effective binder pocket, which is the underlying theory underpinning the Cohen model. It is possible to estimate the amount of agglomerated aluminum powder on composite propellant surfaces using this model. Grigorev's mathematical model can be utilized to calculate some relevant pocket data, including the size distribution function and the parameters of the agglomerates formed by the pocket. As for the model of Babuk [15], the "pocket" and "inter pocket" mechanisms of agglomeration are considered, as well as how the separation of agglomerating metal particles on the burning surface affects the size of agglomerates. A difference of less than 15% was found between the model calculation and the experimental results. Recently, Gallier [16] improved the pocket model by proposing a stochastic pocket model. The biggest advantage of this recently proposed model is that it can derive relevant geometric information from the numerical random pack of the propellant. As far as agglomeration models are concerned, the packing-based model, first developed by Jackson et al. [17], is the most promising, since it can provide all information regarding aluminum particle size. Jackson's model is in reference to a packing algorithm and the agglomeration neighborhood concept. If the distance between two adjacent particles is less than a specified value, the particles will agglomerate. Nevertheless, the fine calibration of the distance defined limits the prediction ability of the model [18]. Muravyev et al [19] studied the morphology, thermal behavior, chemical purity and combustion parameters of HMX as a single-component propellant and aluminum /HMX as a binary system using particles of various sizes, and examined how particle size and microstructure of the components affect the burning rate of energetic systems. According to the results, replacing the micron aluminum powder with an ultrafine aluminum powder can increase combustion speed by 2.5 times and combustion completeness by 4 times. Glotov et al [20]. studied the combustion characteristics of aluminum powder propellants containing AP, HMX, energetic binders, and different polymers under pressures of 0.15 MPa and 4.6 MPa, and concluded that aluminum coated with fluorine reduces agglomeration. Due to the complexity of aluminum agglomeration, a reliable prediction model is still lacking.

CCPs are formed through both agglomeration and combustion [21,22]. Despite the agglomeration models mentioned above, few studies have attempted to develop a prediction model for CCPs. Babuk [23] has made pioneering contributions by proposing a mathematical model of CCP formation and evolution in the motor chamber. A detailed discussion was made of the physico-chemical transformation, particle size, chemical composition, and structure of the particles. According to Jackson's model, the agglomerate size distribution near the burning surface can be described in terms of three-dimensional propellant structure packing, but it ignores the influence of the combustion process on the evolution of the agglomerate. Furthermore, all these prediction models have been validated at low pressures under 7 MPa, while no relevant research has been conducted to reflect the actual solid rocket motor operating conditions. Herein, we present a new analytical model for predicting the size distribution of CCPs of solid propellants. We first develop an aluminum agglomeration model based on the discrete element method, using parameters related to the burning rate and propellant formulation. Then, coupled with a simple aluminum combustion model, the particle-size distribution of CCPs after complete combustion is obtained. The accuracy of the model is verified experimentally by collecting CCPs near and away from the burning surface under 6–10 MPa. The proposed model is expected to be a quick and effective solution for predicting the CCPs of solid propellants.

## 2. Experimental and Numerical Methods

### 2.1. Experimental Method

A constant-pressure quench vessel is used to collect condensed combustion products. The diagram of the experimental setup is shown in Figure 1a, consisting of a cylindrical steel tube measuring 195 mm in diameter, and a thick-walled cylindrical steel chamber measuring 1180 mm in length. It is first necessary to fill the pressure chamber with water and then pressurize it with nitrogen to achieve the required pressure, before lighting the propellant specimen. Moreover, an appropriate trigger time is also set for the ignition valve, drain valve, and pneumatic control valve, in order to achieve the desired results. Prior to burning, the propellant sticks to the bottom of the telescopic rod. As the propellant burns, the telescopic rod moves downward at the same rate as the propellant's burning rate, ensuring that the quench distance is always 2 mm or 40 mm. As the combustion products leave the burning surface, they move downward under the influence of gravity, and eventually condense in the water of the collection device. Using this method, particles near and far from the combustion surface are collected. During the combustion of the propellant, a constant pressure must be maintained by the exhaust liquids or gases. After the propellant has been combusted, the pneumatic control valve closes. Exhaust gases are released after about thirty minutes, by opening an electro-magnetic valve. As the pressure inside the container decreases, the air pressure in the container will reach atmospheric pressure. Afterwards, the liquid quench, along with the slurry (condensed combustion products), is transferred to barrels.

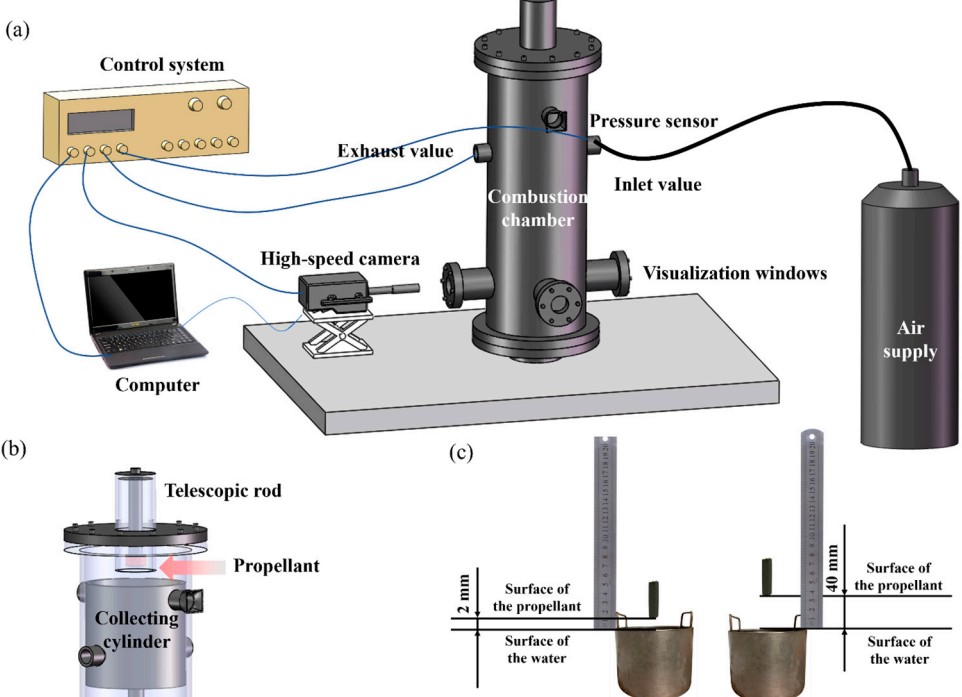

**Figure 1.** Constant-pressure CCP collection system: (**a**) quench vessel; (**b**) propellant with a height controlled by the expansive link and a combustion product collecting device with a cooling medium; (**c**) different quench distances used to determine CCPs.

Experiments are conducted based on HTPB propellants. This study uses a fixed formulation of 11 wt.% hydroxyl-terminated Polybutadiene (HTPB), 70 wt.% AP with a particle diameter of 60–80, 1.5 wt.% catocene (GFP), and 17.5 wt.% Al and dioctyl sebacate (DOS) as the plasticizer. Vinyl resin adhesive is applied to the side of each sample to sustain end burning. Figure 1b illustrates how the propellant is suspended above the collecting cylinder by means of the telescopic rod. A comparison of the collection distances used in the experiment is shown in Figure 1c.

The suspension needs to settle for 48 h in a barrel with a water depth of 20 cm. Based on computational results, 48 h is sufficient for the majority of the suspended particles larger than 1 μm to settle. A 500 mL portion of the mixed liquid is collected into beakers at the bottom of the barrel. To remove impurities, collections are washed several times with ethanol. The solids in the collections are then centrifuged to separate them. Solids are dried under vacuum for 24 h at 70 °C before analysis. A minimum of two tests are performed for each operating condition, to ensure reproducibility. A repeatability of 5% of CCP size distributions, based on the mean-mass diameters of different runs, is observed.

Table 1 presents the experimental schemes. The quench distance is defined as the distance between the surface of the propellant and the surface of the water before the propellant burns. A 0.2 mm/s uncertainty is estimated for the measurement of burning rates.

**Table 1.** Experiments with different quenching distances and chamber pressures.

| Experiment No. | Chamber Pressure/MPa | Burning Rate/mm/s | Quench Distance/mm |
|---|---|---|---|
| 1 | 6 | 4.5 | 2 |
| 2 | 8 | 4.8 | 2 |
| 3 | 10 | 5.1 | 2 |
| 4 | 6 | 4.5 | 40 |
| 5 | 8 | 4.8 | 40 |
| 6 | 10 | 5.1 | 40 |

A laser diffraction particle size analyzer (Malvern Master sizer 2000) is used to measure the particle-size distribution of CCPs directly. Measurement of particle-size distribution requires a sample quality of about 0.1 g. Data on particle-size distribution is obtained by keeping the obscuration between 10% and 20%. The size range is from 0.02 μm to 2000 μm. Using a scanning electron microscope (SEM), powder samples can be viewed at high magnification.

*2.2. Numerical Method*

Based on the distribution of AP particles and aluminum particles within aluminum-containing solid propellants, a geometric topological model is constructed, using the particle filling algorithm. Figure 2a shows the calculation process. By updating the flow field according to the gas-phase combustion products, the position and velocity of aluminum particles above the combustion surface can be solved by a discrete element method. The force of the Al particles on the flow field is added to the gas-phase flow field equations as a source term to achieve fluid–solid coupling. Particle contact is detected when updating the position of the Al particles, and the aggregation rate is updated if aggregation occurs. As time passes, the above process is repeated until the solid propellant has been completely consumed. As shown in Figure 2b, the calculation domain is a three-dimensional cube, with the lower part representing the solid propellant region, the upper part being the agglomerate airflow region, and the interface being the burning surface. A random distribution of aluminum particles is generated in the propellant, according to the particle-size distribution provided. If the particles overlap, the particle position is adjusted by using a viscous suspension method. The upper boundary surface is the outlet, the particles escape and disappear when they leave the boundary, and there are periodic boundary conditions around them. When the burning surface moves down to the bottom surface, the propellant length becomes zero, and the calculation is completed.

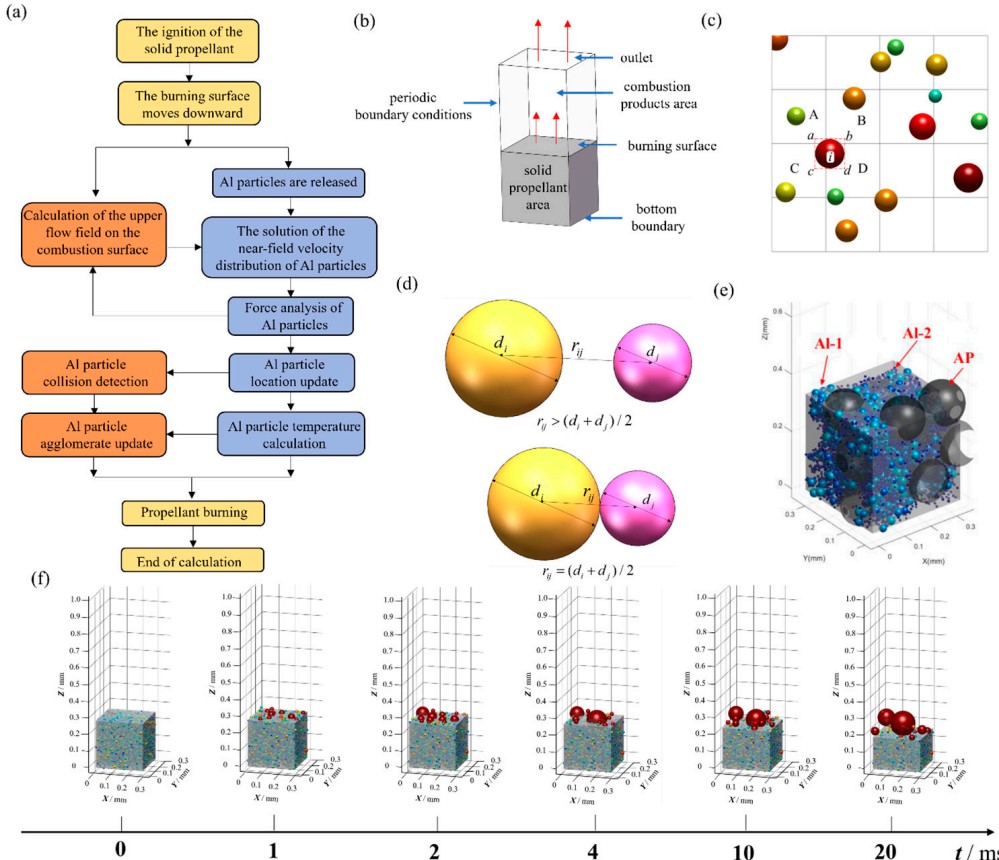

**Figure 2.** Numerical simulation method: (**a**) schematic diagram of the calculation process for Al particle agglomeration; (**b**) calculation domain division; (**c**) aluminum particle contact detection algorithm; (**d**) schematic diagram of the aluminum particle agglomeration criterion; (**e**) initial fine geometric topology of an Al-containing solid propellant; (**f**) agglomeration of Al in a solid propellant.

As a simulation method, discrete element modeling (DEM) relies on direct tracking of the detailed motion of individual particles. It was developed by the American scholar Cundall [24] and applied initially to geotechnical mechanics. The basic idea of DEM is to decompose an object into many small discrete elements, and then simulate the behavior of the object by calculating the interaction between these elements. The steps of DEM include discretization, establishing an interaction model, solving, and analyzing. It is first necessary to decompose the object into small discrete elements, such as spheres, cubes, or other shapes. Then, a model of the interaction between the elements is developed, including elasticity, friction, collisions, etc. The next step is to solve the interaction force and the motion equation between elements, in order to simulate the behavior of the object. As a final step, the simulation results should be analyzed and verified, in order to determine the accuracy and reliability of the model. During DEM simulation, the calculation of particle contact detection is very time-consuming. Using an efficient grid detection method, the process is sped up in this study. Figure 2c shows that the computational domain consists of rectangular grids larger than the particle size. Particles are represented by different spheres. There is no special significance to spheres of different colors. Each particle occupies up to four grids in two dimensions. Taking particle i as an example, according to its four vertices A, B, C and D, the grids adjacent to i, namely A, B, C and D, can be obtained. Particle i can only contact the particles contained in these four grids, so the detection range can be narrowed. The method is similar for a three-dimensional situation. Only the particles in up to eight grids occupied by particle i need to be detected. Traditional DEM calculations usually use a cell size of 1 to 2 times the particle diameter. However, it is not necessary to use such a small unit size for aerosols, since they are usually diluted and have a very

low volume fraction. Searching neighboring particles may lead to unnecessary calculation and increase the calculation time. According to our findings, the optimal time step in the minimum calculation time is basically equal to the initial total number of aerosol particles in the calculation domain. The motion equations of a single aluminum particle determined by the forces were [25]:

$$\frac{\partial^2 X}{\partial t^2} = \frac{1}{m_{\text{p}}}\left(\sum F_{\text{i},x} + \sum F_{\text{D},x}\right) \tag{1}$$

$$\frac{\partial^2 Y}{\partial t^2} = \frac{1}{m_{\text{p}}}\left(\sum F_{\text{i},y} + \sum F_{\text{D},y}\right) \tag{2}$$

$$\frac{\partial^2 Z}{\partial t^2} = \frac{1}{m_{\text{p}}}\left(\sum F_{\text{i},z} + \sum F_{\text{D},z} + F_{\text{g}}\right) \tag{3}$$

where $X$, $Y$, and $Z$ denote the spatial position coordinates of the aluminum particles; $m_{\text{p}}$ is the currently calculated mass of the aluminum particle; $\sum F_{\text{i}}$ is the contact force exerted on the aluminum particle by other aluminum particles; $\sum F_{\text{D}}$ is the drag force on the current aluminum particle; and $\sum F_{\text{g}}$ is the gravitational force on the current aluminum particle (assuming gravity is in the $z$-axis direction).

As shown in Figure 2d, when the distance between particles i and j meets the following condition, agglomeration is considered to occur:

$$r_{\text{ij}} \leq \frac{d_{\text{i}} + d_{\text{j}}}{2} \tag{4}$$

where $r_{\text{ij}}$ is the distance between the centers of aluminum particles i and j, and $d_{\text{i}}$ and $d_{\text{j}}$ are the diameters of aluminum particles i and j, respectively.

After the aluminum particles i and j agglomerate, the corresponding velocities change:

$$u_{\text{a}} = \frac{m_{\text{i}}u_{\text{i}} + m_{\text{j}}u_{\text{j}}}{m_{\text{i}} + m_{\text{j}}} \tag{5}$$

where $u_{\text{a}}$ is the velocity of motion after agglomeration; $m_{\text{i}}$ and $m_{\text{j}}$ are the masses of aluminum particles i and j, respectively; and $u_{\text{i}}$ and $u_{\text{j}}$ are the velocities of aluminum particles i and j, respectively.

There is a compact structure to solid propellants that contain aluminum. A continuous phase matrix composed of HTPB disperses Al and AP particles of different sizes, freely. Figure 2b illustrates the calculation domain that has been set after which AP and Al particles are randomly placed in the area of a solid propellant containing aluminum, to simulate the fine geometry. In spite of the non-spherical shape of propellant particulates (especially AP particles), they are uniformly represented as spherical particles for the purpose of simplifying the analysis and calculation. A viscous suspension method (VSM) is used to maximize particle filling efficiency in this study and an ensemble rearrangement is used for determining particle spatial positions [23]. Figure 2e shows the geometric topology of solid propellant constructed by the above method. As shown in the diagram, AP particles appear as large black balls, and Al particles appear as small particles dispersed among the large ones. Al-1 and Al-2 stand for different Al particles. An aluminum particle aggregation simulation is then based on this topology.

Figure 2f shows the typical simulation results of aluminum agglomeration. As the burning surface gradually retreats, the aluminum particles that initially overflowed from the propellant agglomerate on it. The initial stage of agglomeration occurs over the first 1 to 2 ms, when the size of agglomerates is small, and agglomerates are formed on the burning surface. As agglomeration persists, the particles enter the middle stage, which lasts from 4 to 10 ms, the middle stage of agglomeration. After 40 ms of agglomeration, the agglomerates leave the burning surface and reach their largest size.

## 3. Results and Discussions

### 3.1. Agglomerate Size Distribution near the Burning Surface

The SEM overview of typical CCPs collected near the burning surface is shown in Figure 3a. Although some of the particles have been ignited, they can still be considered unprocessed agglomerates that can be used to validate the particle-size distribution produced by the agglomeration model. The CCPs range in size from 0.3 μm to 600 μm, with a multimodal distribution. The three size peaks are 1 μm~2 μm, 20 μm~30 μm, and over 300 μm. In general, soot particles and agglomerates are considered to constitute CCPs. The particle size of alumina soot is about 1 μm, and the agglomerate size can usually reach hundreds of microns. The particles of 1~2 μm are composed of alumina soot, which is formed as a result of the oxidation of aluminum vapor. During combustion, aluminum particles agglomerate or form molten oxide shells, producing residual oxide particles between 20 μm and 30 μm. Finally, particles larger than 300 μm are considered large aggregates. It is consistent with Ao's [2] and Jeenu's [26] work that the particles of the present study have a trimodal distribution of particle size.

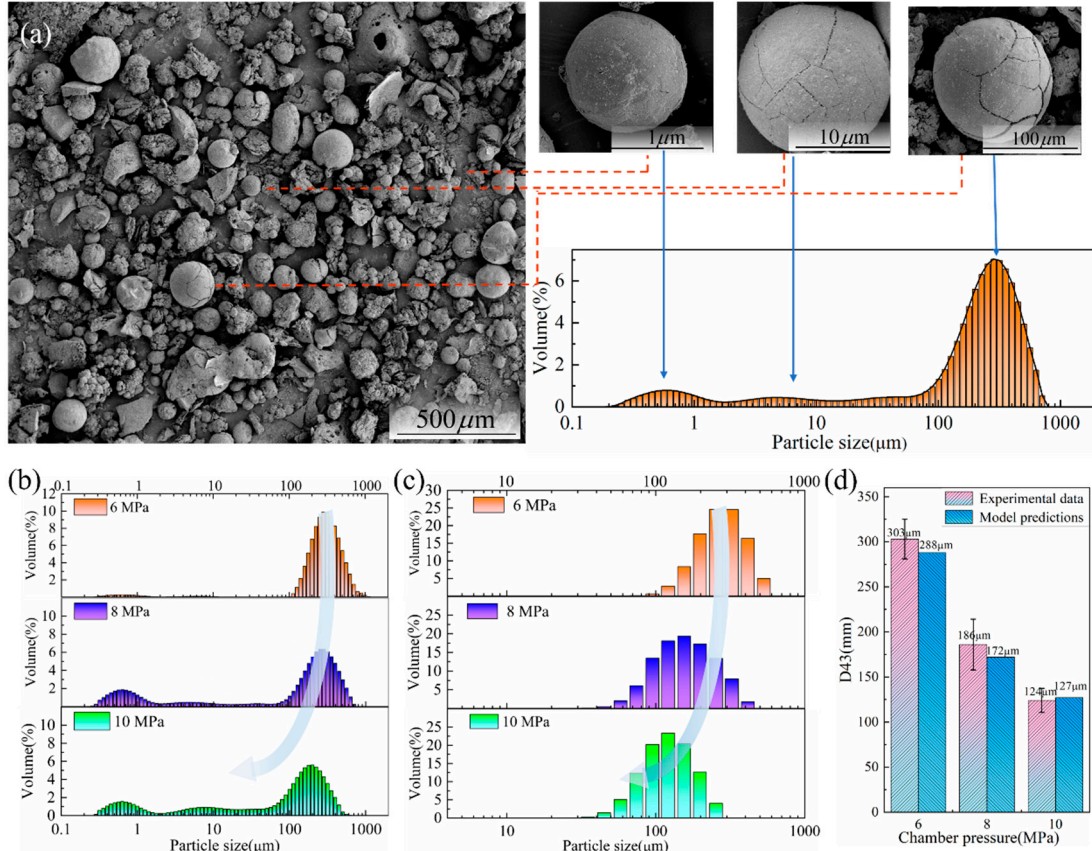

**Figure 3.** High-pressure verification of the aluminum agglomeration model; (**a**) SEM overview images of typical condensed combustion products show agglomerates and smoke oxide particles distributed in the products; (**b**) experimental particle-size distribution near the burning surface; (**c**) numerical calculation of agglomerate size distribution near the burning surface; (**d**) comparison between experimental and numerical $D_{43}$ data near the burning surface.

Figure 3b shows the size distribution of CCPs near the burning surface, under different pressures. As stated before, there is a multimodal distribution of particle sizes. When the chamber pressure is 6 MPa, the proportion of 283-μm agglomerates reaches the highest, 9.9% of the total volume. Most particles have a size between 100 μm and 1125 μm, accounting for 95.9% of the total volume. When the chamber pressure is 8 MPa, the agglomerate peak still accounts for highest content, and most of the particles are distributed between 70 μm and

710 μm. When the pressure reaches 10 MPa, the agglomerate peak moves further left, to 200 μm, accounting for 64.9% of the total volume, while the soot and residual peaks begin to increase. For a more accurate description of agglomerate particle size characteristics, we introduce equivalent particle size $D_{43}$, which refers to area of the average particle size. $D_{43}$ is the mass-averaged diameter of the agglomerates and is calculated as $D_{43} = \sum D_i^4 / \sum D_i^3$ Ref. [27]. The mean diameter of the agglomerates $D_{43}$ for 6 MPa, 8 MPa and 10 MPa are 303 μm, 186 μm, and 124 μm, respectively. It is obvious that agglomerate sizes decrease with increasing pressure. As pressure increases, it is evident that the burning rate increases as well. The burning rate may reduce agglomerate size, since the velocity of the burning gas near the burning surface increases, removing the agglomerates more quickly [28]. Steam combustion of aluminum produces smoke oxide particles (SOPs). SOPs become smaller with pressure, as their mean diameter decreases.

Figure 3c shows the size distribution results for CCPs near the burning surface at different chamber pressures, by numerical calculating. It should be noted that SOPs and residual oxides generated by combustion are not taken into account. The results show that the agglomerate peak shifts from 254 μm to 121 μm as the chamber pressure increases from 6 MPa to 10 MPa. The experimental data is compared with the simulated $D_{43}$ results, as shown in Figure 3d. Compared with the experimental data, the simulation results are remarkably similar. The deviation is only 5.0% at a chamber pressure of 6 MPa, 7.5% at 8 MPa, and 2.4% at 10 MPa, all less than 10%. Therefore, the current model basically predicts the size of aluminum agglomerates near the burning surface of solid propellants during combustion.

### 3.2. Analytical Model for CCP Size Prediction

As CCPs evolve in the multi-phase flow, they form the final propellant combustion products. CCPs are formed by agglomerates on the surface layer of burning propellants. There are usually two fractions of particle sizes and properties present in CCPs, i.e., fine alumina (FA) and agglomerates. The production of FA results from a variety of processes, including the condensation of aluminum gas-phase (vapor-phase) combustion products, the condensation of gas-phase combustion products of unagglomerated metal particles, and the heterogeneous combustion of unagglomerated metal particles [15]. In contrast, large CCPs are produced by the combustion residuals of agglomerates. Figure 4a shows the typical evolution of CCPs.

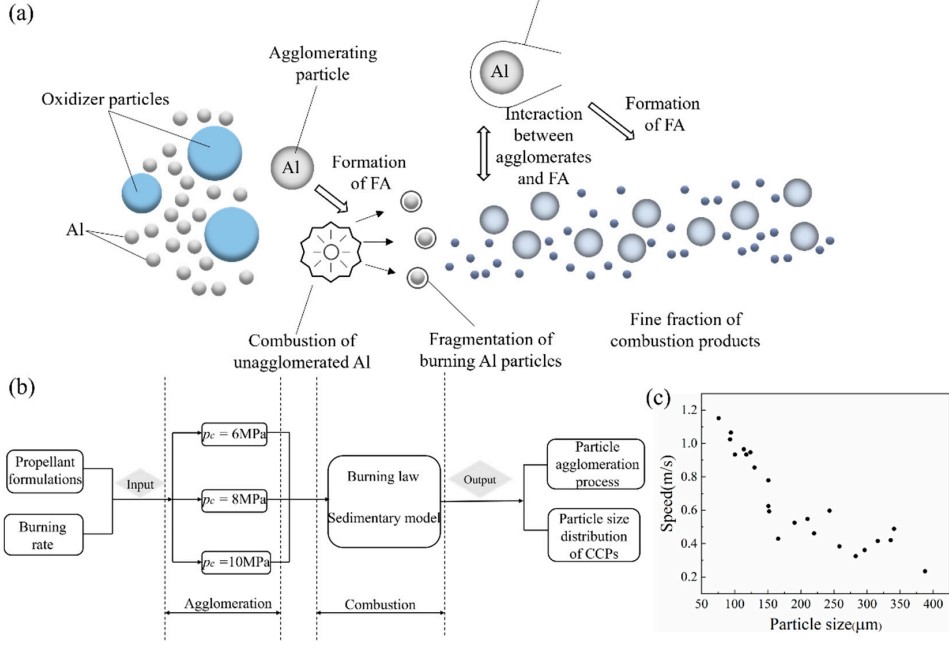

**Figure 4.** Analytical model of CCP size prediction: (**a**) schematic of the evolution of CCPs; (**b**) scheme of particle size prediction of CCPs; (**c**) velocity of particles with different sizes in the propellant plume.

According to the previous method of calculation, the process for calculating CCP size can be derived, as shown in Figure 4b. The input contents are propellant formula, burning rate, and ambient pressure. After calculating the agglomeration under different pressures, the aluminum particles are examined for their combustion process. The final output is the particle-size distribution of CCPs. The agglomeration process is calculated using the DEM method, while the combustion process is mainly calculated using the combustion model and deposition model. According to Melcher's work, the combustion law of aluminum particles in a solid rocket motor chamber accords with [29]:

$$D = D_0 - 20t \tag{6}$$

$D$ denotes the diameter of the aluminum droplet, and $D_0$ the diameter of the initial aluminum droplet. The unit of $t$ is the millisecond. The deposition rate is defined as the ratio between the metal oxide mass and the initial particle mass. Assuming that all particles are spherical, the deposition rate can be expressed as:

$$\beta = \frac{m_{mox}}{m_{P_0}} = \frac{\sum_{i=1}^{n_{Al_2O_3}} \rho_{Al_2O_3} N_{Al_2O_3,i} \frac{\pi}{6} D^3_{Al_2O_3,i}}{\sum_{i=1}^{n_{Al}} \rho_{Al} N_{Al,i} \frac{\pi}{6} D^3_{Al,i}} \tag{7}$$

where $m_{mox}$ is the metal oxide mass, $m_{P_0}$ is the initial particle mass, $\rho_{Al_2O_3}$ is the alumina density, $N_{Al_2O_3,i}$ is the number of alumina particles of the current size, $D_{Al_2O_3,i}$ is the diameter of the alumina particles, $n_{Al_2O_3}$ is the total number of alumina particle sizes, $\rho_{Al}$ is the aluminum density, $N_{Al,i}$ is the number of alumina particles with the current particle size, $D_{Al,i}$ is the diameter of the alumina particles, and $n_A$ is the total number of alumina particle sizes. The experimental results for particle velocities as they separate from the burning surface are shown in Figure 4c. As particle size increases, the velocity of particles in gas decreases. The velocity of all particles is less than 1.2 m/s. There is a 40 mm quenching distance between the liquid surface and the propellant surface during CCP collection. Combined with the particle speed, it is calculated that aluminum burns completely when particles reach the liquid surface. Therefore, the CCPs collected above 40 mm can be confidently identified as alumina. Moreover, the alumina density is 3900 kg/m$^3$ and the aluminum density is 2700 kg/m$^3$. According to King et al. [30], the particle size of aluminum particles after combustion in solid propellants is about 70% of that before combustion. On this basis, using the calculation Formula (3), the deposition rate is 0.50. Meanwhile, according to DesJardin et al. [31], the deposition rate of propellants in combustion environments is about 50%. Therefore, it is assumed that the deposition rate at pressures of 6~10 MPa is 0.50 in this study. The residue size after combustion can thus be determined.

### 3.3. Experimental Verification

The overall morphology of the CCPs collected near and away from the burning surface under different pressures is shown in Figure 5a. According to the comparison of microstructures, there are two significant differences between the aggregates at two locations. First, as a result of combustion, agglomerates that are situated farther from the burning surface have a smaller particle size in general. As mentioned above, most of the agglomerates near the burning surface have just escaped from the burning surface and have just ignited, so the composition is primarily aluminum. The particles away from the burning surface, however, have been fully burned, and the aluminum has become alumina. Second, there is a large number of metal flocs in the particles near the burning surface, originating from the agglomerates that have formed during the agglomeration process. The characteristics of metal flocs have been discussed by a number of researchers [28]. These flocs almost disappear at the 40 mm position. Most of the agglomerate droplets are spherical, due to the surface tension during combustion, so these smooth spherical particles

can represent the results of combustion. To summarize, the evolution from agglomeration to combustion can be observed by collecting particles from different locations.

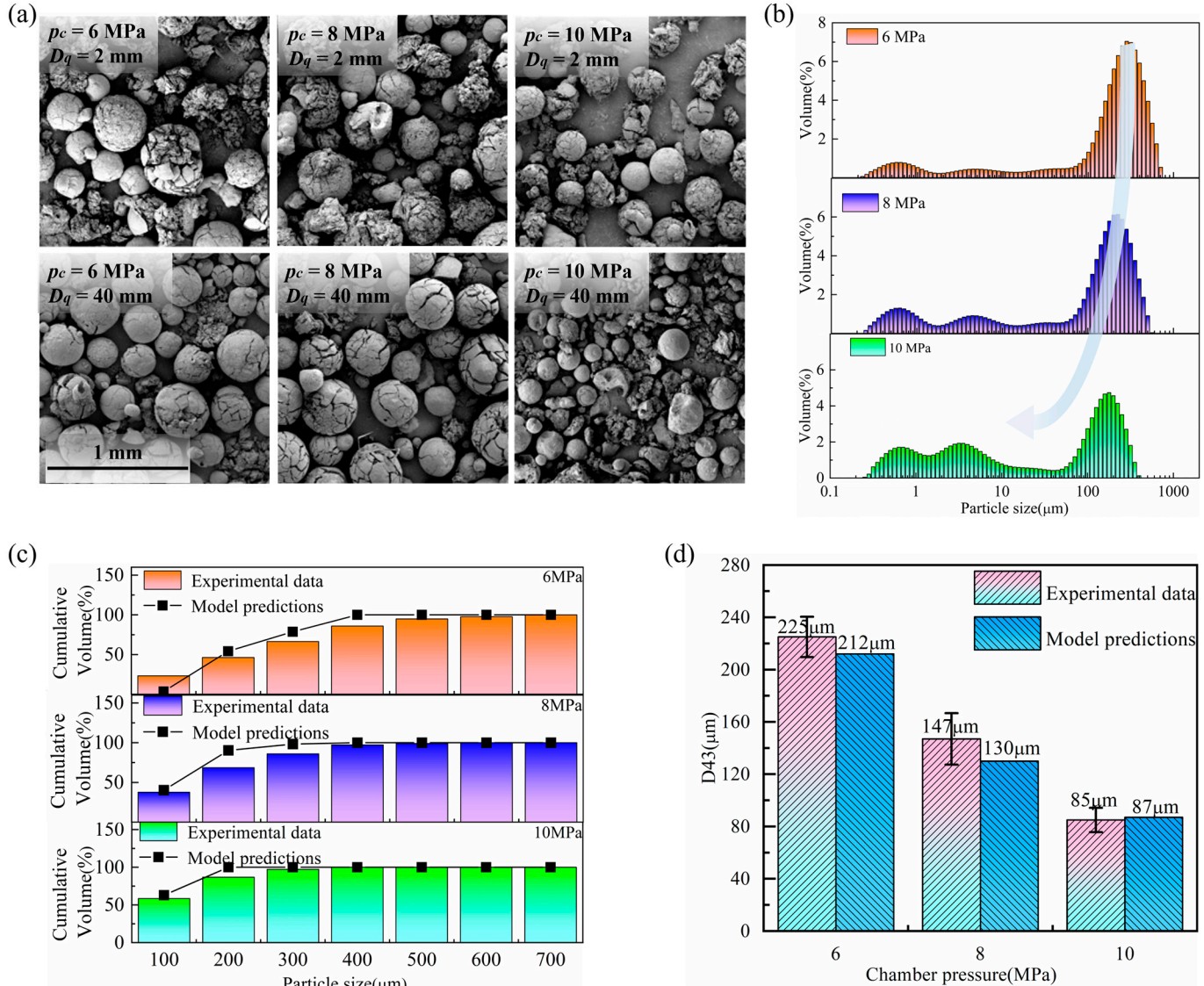

**Figure 5.** Experimental verification of CCPs evolution: (**a**) overall morphology of CCPs, showing a decreasing trend of particle size before and after combustion and the size distribution of CCPs away from the burning surface at different pressures; (**b**) experimental particle-size distribution away from the burning surface; (**c**) the cumulative fraction by particle size of experimental data and model predictions; (**d**) comparison between experimental and numerical $D_{43}$ of alumina away from the burning surface.

Figure 5b shows CCPs size distributions away from the burning surface. The particle-size distribution is similar to that of the initial agglomerates, which is also a multimodal distribution. When the chamber pressure is 6 MPa, 8 MPa, and 10 MPa, the particle sizes with the highest content of CCPs are 283 μm, 224 μm, and 178 μm, respectively. The mean diameter of agglomerates near the burning surface, $D_{43}$, is 221 μm, 142 μm, and 81 μm, respectively, for pressures of 6 MPa, 8 MPa, and 10 MPa. As the quench distance increases, the agglomerate size decreases. $D_{43}$ is reduced to 82 μm at a chamber pressure of 6 MPa, 44 μm at 8 MPa, and 43 μm at 10 MPa. The variation trend of $D_{43}$ with quench distance is consistent with that shown in SEM images.

As shown in Figure 5c, both experimental data and model predictions show similar trends in particle-size distribution. The experimental $D_{43}$ of alumina is compared with the simulated results, as shown in Figure 5d. The experimental and theoretical results are in good agreement. The deviation is 4.1%, 8.5%, and 7.4% at 6 MPa, 8 MPa and 10 MPa, respectively. Thus, the present analytical model can accurately predict both the particle-size distribution and average agglomerate diameter, which plays a critical role in CCPs. Although the particle-size distribution of SOPs is not predicted in the model, large agglomerates are the key to determining two-phase flow loss and ablation. In the future, more sophisticated particle combustion models should be developed, so as to fully depict the complete particle field.

## 4. Conclusions

A numerical simulation was performed using the DEM-based numerical simulation approach to investigate aluminum particle agglomeration during solid propellant combustion. The simulation method takes into account precipitation, particle traction and, turbulence effects, as well as heating and melting of aluminum particles. A grid-based method of contact detection was used to accelerate the calculations. An analytical model for predicting CCPs sizes was also proposed by combining the combustion and deposition model of aluminum particles. A CCP collection device was used to collect metal particles near and away from the burning surface under high pressure (6~10 MPa), representing the initial agglomerates and final CCPs, respectively. The experimental data was used to verify the proposed CCPs prediction model. The calculations and experiments were in good agreement, with an error ranging between 4.1 and 8.5%, validating the effectiveness and accuracy of the model. In the future, the size distribution of SOPs needs to be considered in the model, to make it more comprehensive.

**Author Contributions:** Conceptualization, S.Y. and H.L.; methodology, W.A.; software, L.L.; validation, S.Y., H.L. and Y.J.; formal analysis, L.L.; investigation, S.Y.; resources, H.L.; data curation, W.A. and A.P.; writing—original draft preparation, S.Y., Y.J. and L.L.; writing—review and editing, H.L., W.A., P.L., A.P. and G.Z.; funding acquisition, P.L. All authors have read and agreed to the published version of the manuscript.

**Funding:** This research was funded by National Natural Science Foundation of China grant No. 21975066 and No. U2241250.

**Data Availability Statement:** The data presented in this study are available on request from the corresponding author. The data are not publicly available due to due to restrictions privacy.

**Conflicts of Interest:** The authors declare no conflict of interest.

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
