# Peer review of "Agglomerate Size Evolution in Solid Propellant Combustion under High Pressure"

_aerospace, doi:10.3390/aerospace10060515_

Round 1

Author Response

Reviewer #2: For the average scientific reader, this paper is incomprehensible. I list the following confusing elements as examples:

Our response: Thank you very much for taking the time to review our manuscript. Please find below our response to your comments, which we found to be very helpful. Changes made in response to the comments are highlighted in blue in our revised manuscript.

1) Nowhere could I find the definition of the abbreviation “AP’.

Our response: In this work, AP means ammonium perchlorate. We have added this information to our revised manuscript.

2) On page 2, the work of someone named “Cohen” is mentioned but none of the references contains this name among the authors.

Our response: Thanks for the problems found by the reviewers! We supplemented the reference [13], which is Cohen's research results.

3) Part © of Fig. 1 is so poorly labeled as to be incomprehensible.

Our response: Fig.1(c) shows two quench distances in the experiment, which is the distance between the burning surface of two propellants and the water surface. The figure has been enhanced with supplementary marks to facilitate the reader's understanding.

4) The print in Fig. 2 is so small. Even a standard power magnifying glass fails to reveal the lettering.

Our response: We appreciate the suggestions from the reviewers! The text in Fig.2 has been enlarged in our revised manuscript.

5) Page 5. What is a DEM simulation?

Our response: As a simulation method, discrete element modeling (DEM) relies on direct tracking of the detailed motion of individual particles. The basic idea of DEM is to decompose an object into many small discrete elements, and then simulate the behavior of the object by calculating the interaction between these elements. The steps of DEM include discretization, establishing interaction model, solving and analyzing. It is first necessary to decompose the object into small discrete elements, such as spheres, cubes, or other shapes. Then, a model of the interaction between elements is developed, including elasticity, friction, collisions, etc. The next step is to solve the interaction force and motion equation between elements in order to simulate the behavior of the object. As a final step, the simulation results should be analyzed and verified in order to determine the accuracy and reliability of the model. The method is capable of considering particle-particle and airflow-particle interactions at the microscopic level, and it has been explored in recent years for the simulation of particle agglomeration. We have added this information to our revised manuscript.

6) Page 6. Eqs.(1) – (3) are apparently Newton’s law of motion applied to particle motion in three dimensions. In the text below these equations, is used to represent the gravitational forces, but it appears nowhere in the equations of motion.

Our response: Thanks for the problems found by the reviewer. The original text contained an error in Eqs. (3) and has now been corrected.

7) The authors point out that they are not relying on numerical simulation, but in fact, other than Eqs.(1) – (3), they are doing just that.

Our response: Our apologies for causing the reviewers to misunderstand. The paper is not independent of numerical simulation. On the contrary, the purpose of this paper is to present an analysis model of the evolution of CCPs and to verify its applicability through experiments. Therefore, numerical simulation is an important component of this study.

8) Page 8. Why should the mean diameter symbol, D43, be given the subscript, “43”?

Our response: As mentioned in many references, D43 is the mass-averaged diameter of the agglomerates and is calculated as D43=∑Di4/∑Di3. (Ref.[27]). We have added this information to our revised manuscript.

Ref.[27] Babuk, V. A.; Vassiliev, V. A.; Sviridov, V. V.. Propellant formulation factors and metal agglomeration in combustion of aluminized solid rocket propellant. J. Combustion Science & Technology, 2001, 163, 261-289. [CrossRef]

9) Page 8. The work of Ao and Jeenu is mentioned, but there is no citation in the text.

Our response: References have been supplemented and Ao and Jeenu's work has been cited.

The idea of the paper is not to summarize the work for the benefit of the authors, who already are familiar with the methods and the jargon, but rather to instruct the reader sufficiently that he is convinced that there is something worth publishing. This paper needs to be rewritten in toto.

Our response: Thank you very much for your valuable advice. We have made three revisions to enhance the readability of the manuscript.

Firstly, the introduction part has been rewritten. We point out that there are few studies trying to develop the prediction model of CCP at present, especially under the high pressures which is more similar to the actual working conditions of solid rocket motor.

Secondly, the methods and terms are introduced in more detail. The definitions of some terms such as AP and D43 are supplemented, and the principle and steps of DEM are introduced in detail.

In addition, the structure of the paper has been rearranged ("3.2 aggregated size distribution away from the burning surface" has been changed to "analytical model for CCPs size prediction"; Change "3.3 numerical calculation of CCPs size evolution" to "3.3 Experimental Verification" to make it easier for readers to understand. Changes made in response to the comments are highlighted in blue in our revised manuscript.

Reviewer 2 Report

It is an interesting contribution to the modeling of aluminum agglomeration in the course of combustion of solid propellants. So, i think that the research willbe of interest to readers of the journal. I have no serious comments, just recommend the authors to describe more extensively the core model. Also, in the introduction section the following research can be discussed - https://doi.org/10.1002/prep.201000028

https://doi.org/10.1007/s10573-007-0045-y

- to show the prospect of reduction of the metal parcticle size and other peculiarities of Al agglomeration process

Author Response

Thank you very much for taking the time to review our manuscript. We have carefully studied these two research results and discussed them in the introduction:

Muravyev[19] et al. studied the morphology, thermal behavior, chemical purity and combustion parameters of HMX as a single-component propellant and aluminum /HMX as a binary system using particles of various sizes, and examined how particle size and microstructure of components affect the burning rate of energetic systems. According to the results, replacing micron aluminum powder with ultrafine aluminum powder can increase combustion speed by 2.5 times and combustion completeness by four times. Glotov[20] et al. studied the combustion characteristics of aluminum powder propellants containing AP, HMX, energetic binders, and different polymers under pressures of 0.15 MPa and 4.6 MPa, and concluded that aluminum coated with fluorine reduces agglomeration.

Ref.[19] Muravyev, N.; Frolov, Y.; Pivkina, A.. Influence of particle size and mixing technology on combustion of HMX/Al compositions. J. Propellants explosives pyrotechnics. 2009, 35, 226-232. [CrossRef]

Ref.[20] Glotov, O. G.; Yagodnikov, D. A.; Vorobev, V. S.. Ignition, combustion, and agglomeration of encapsulated aluminum particles in a composite solid propellant. II. Experimental studies of agglomeration. J. Combustion Explosion & Shock Waves, 2007, 43, 320-333. [CrossRef]

Round 2

Author Response

Response to Reviewer 1

Reviewer #2: The revised manuscript is improved but is still confusing in places. Please see the list below:

Our response: Thank you very much for taking the time to review our manuscript. Please find below our response to your comments, which we found to be very helpful. Changes made in response to the comments are highlighted in blue in our revised manuscript.

1) The experimental procedure described in the text and illustrated in Fig. 1 is confusing. I'll take a guess. The propellant is somehow contained and allowed to burn over a container of water. In one experiment, the distance from the top surface of the water to the bottom surface of the propellant is 2 mm and, in another experiment, it is 40 mm. If my surmise is correct, the heat from the burning propellant vaporizes the water, which in turn extinguishes the combustion. If this is the case, the propellant burns for a longer time at the 40mm. separation than at the 2 mm. separation. If this is so, how as claimed in the text, are the particles near and far from the burning surface separated and collected? Still part (b) of Fig. 1 is obscure. What are roll is played by the expansion link and by the collecting device and how do they function?

Our response: Our apologies for causing the reviewers to misunderstand. Due to the fact that the propellant is not extinguished by evaporated water, but by natural extinguishment after combustion, the combustion time remains the same regardless of the environment.

Prior to burning, the propellant sticks to the bottom of the telescopic rod. As the propellant burns, the telescopic rod moves downward at the same rate as the propellant's burning rate, ensuring that the quench distance is always 2 mm or 40 mm. As the combustion products leave the burning surface, they move downward under the influence of gravity, and eventually condense in the water of the collection device. Using this method, particles near and far from the combustion surface are collected.

We have added this information to our revised manuscript.

2) Section 2.1 Experimental Method. The terms used in the text do not always agree with terms used to designate the various devices in Fig.1.

Our response: Thanks for the problems found by the reviewers! Fig.1 has been modified to ensure that the terms of various devices are consistent with those used in the text.

3) Section 2.2. First paragraph mentions a source term in the gas-phase flow field equation. Source of what? Gas? Al particles?

Our response: Our apologies for not making it clear in the text. The force of the Al particles on the flow field is added to the gas-phase flow field equations as a source term to achieve fluid-solid coupling. In our revised manuscript, the sentence has been rephrased in a clearer manner.

4) Fig.2. Part c. We are not told what distinguishes the yellow and green spheres.

Our response: Fig.2. Part c represents the aluminum particle contact detection algorithm, which is a schematic diagram of the grid detection algorithm. Particles are represented by different spheres. There is no special significance to spheres of different colors. We have added this information to our revised manuscript.

5) Fig.2.Part (f) is not labeled as such. Rather it also has the label, (e), as does the preceding part. Here is what I glean from what should be part (f). The aluminum particles start out dispersed uniformly throughout the propellant. As the propellant burns at the top and its surface retreats, aluminum oxide agglomerates are left behind on the surface, which retreats.

Our response: Thanks for the problems found by the reviewer. The label in Figure 2 is wrong, which has been revised in the manuscript.

6) Fig.2.Part(e). What distinguishes the particles designated by "Al-1" "A1-2"?

Our response: The large black spheres in the figure represent AP particles, and the small particles dispersed between the large spheres represent Al particles. Al-1 and Al-2 stand for different Al particles. We have added this information to our revised manuscript.

7) Fig.2. There is a mysterious curving white stripe that begins on the bottom panes of both parts (b) and (c) and curves upward generally following the maxima of each size distribution. THERE IS NO EXPLANATION. What in the world are the authors trying to communicate?

Our response: Our apologies for causing the reviewers to misunderstand. In Fig.2, some of the content is exported with a background other than white. The revised draft has now modified the picture.

8) Page 10. Second paragraph, first line. The term "psychophysical" is used out of context? Look it up. It is a term used in psychology, not physics.

Our response: Thanks to the reviewers for their valuable comments. We have changed "psychophysical mechanisms" in the article to "method of calculation".

9) Page 11. The symbols,D1,0 and D01,0 in Eq.(6) are nowhere defined in the text. What are the units of t?

Our response: D denotes the diameter of the aluminum droplet, D0 the diameter of the initial aluminum droplet, and 1.0 stands for power 1. The unit of t is millisecond. We have added this information to our revised manuscript. In order to make the formula more understandable to readers, it has been changed to: D=D0-20t.

10) Page 11.The symbol,mP0 , in Eq.(7) is not defined in the text.

Our response: mP0 means the initial particle mass. It is defined in line 7 on page 11 of the original text. To facilitate reading, the definition of mP0 has been moved to the back of Eq.(7).

11) Fig.4. The two sides of part (a) of Fig. 4 seem to tell different things and if so, they should be separated.

Our response: Our apologies for causing the reviewers to misunderstand. Fig. 4(a) illustrates the development process of CCPs as a whole, not as two parts. It has been arranged in a more compact manner to facilitate reader understanding.

12) Page 12. Blue highlighted text. “. .. been discussed by a number of researchers [ref].” Needs a reference.

Our response: Thanks to the suggestions made by the reviewers, references have been added now.

13) Page 13. I could not find a definition of "SOP" anywhere in the text.

Our response: Steam combustion of aluminum produces smoke oxide particles (SOPs). We have added this information to our revised manuscript.

14) The authors have solved Newton's Law of Motion for their particles in the time domain. For a complete solution, some initial conditions must be specified. The authors do not state these.

Our response: The initial condition is that the x, y and z coordinates of particles are all 0. We have added this information to our revised manuscript.

There are some language difficulties, which I list below: Although the pages are numbered, there are no line numbers, so the authors will have to do the best they can in locating the recommended changes in the text.

  1. Page 1. Bottom of the first paragraph. Change "...the CCPs particles..." to read...the CCP particles. .."'
  2. Page 2. First line. Should read "Using these models, the particle size..."3. Page 2. Line 5. Should read, "The pocket model, by . . .""
  3. Page 2. Line 6. Should read, "The basic idea of the pocket model is..."
  4. Page 2. Lines 7 and 8. Should read. The models of Cohen[13] and Grigorev[14]are the most widely known of all pocket models.?'
  5. Page 2. Near the middle of the page. Should read "...as well as how the separation of agglomerating metal particles on the burning surface affects the size of agglomerates.. . ."
  6. Page 2. Near the bottom of the page. Should read.“...specified value, the particles will agglomerate."
  7. Page 2. Just above the blue highlighted material. The following sentence makes no sense." Nevertheless, fine calibration of distance defined limits the prediction ability of the model.”’ Maybe the authors mean to say that a too finely divided mesh over which the differential equations are integrated limits the scope of predictions of the model?
  8. Page 2. Blue highlighted material at the bottom of the page. Last sentence should read, .. . results, replacing the micron aluminum powder with an ultrafine...""

10.Pages 1- 3. The second paragraph of the Introduction runs over three pages. It is much too long. Break it up into the separate ideas which it covers.

  1. Page 3. Second paragraph. Should read, "...is verified experimentally by collecting..."

12.Page 6. Since Eqs. (1) and (2) at the bottom of the page are ordinary differential equations where the independent variable is time, the authors should replace in the line above the term, "optimal units" with the term "optimal time step".

13.Bottom of page 7. Last line. Should read, “The initial stage of agglomeration occurs over the first 1 to 2 ms. ... As agglomeration persists, the particles enter the middle stage which lasts from 4 to 10 ms.

Our response: Thank you very much for your valuable advice. We have corrected the language mistakes as suggested.
